# Effect of Au-Coating on the Laser Spot Cutting on Spring Contact Probe (SCP) for Semi-Conductor Inspection

**DOI:** 10.3390/ma14123300

**Published:** 2021-06-15

**Authors:** Youngjin Seo, Jungsoo Nam, Huitaek Yun, Martin Byung Guk Jun, Dongkyoung Lee

**Affiliations:** 1Department of Future Convergence Engineering, Kongju National University, Cheonan 31080, Korea; syjvlfry1004@gmail.com; 2Intelligent Manufacturing System R&D Department, Korea Institute of Industrial Technology, Cheonan 31056, Korea; rack1219@kitech.re.kr; 3Department of Mechanical Engineering, Purdue University, West Lafayette, IN 47907-2088, USA; yun37@purdue.edu (H.Y.); mbgjun@purdue.edu (M.B.G.J.); 4Department of Mechanical and Automotive Engineering, Kongju National University, Cheonan 31080, Korea

**Keywords:** laser spot cutting, heat-affected zone, material removal zone, Au-coating

## Abstract

Spring contact probes (SCPs) are used to make contact with various test points on printed circuit boards (PCBs), wire harnesses, and connectors. Moreover, they can consist of the test interface between the PCBs and the semiconductor devices. For mass production of SCPs, ultra-small precision components have been manufactured by conventional cutting methods. However, these cutting methods adversely affect the performance of components due to tool wear and extreme shear stress at the contact point. To solve this problem, laser spot cutting is applied to Au-coated SCP specimens as an alternative technique. A 20 W nano-second pulsed Ytterbium fiber laser is used, and the experimental variables are different laser parameters including the pulse duration and repetition rate. After the spot cutting experiments, the heat-affected zone (HAZ) and material removal zone (MRZ) formed by different total irradiated energy (*E_total_*) was observed by using a scanning electron microscope (SEM). Then, the size of HAZ, top and bottom parts of MRZ, and roundness were measured. Furthermore, the change rate of HAZ and MRZ on Au-coated and non-coated specimens was analyzed with regard to different pulse durations. Based on these results, the effect of Au-coating on the SCP was evaluated through the comparison with the non-coated specimen. Consequently, in the Au-coated specimen, hole penetration was observed at a low pulse duration and low total energy due to the higher thermal conductivity of Au. From this study, the applicability of laser spot cutting to Au-coated SCP is investigated.

## 1. Introduction

Various mechanical cutting methods have been widely used to fabricate ultra-small precision components of the electronic device for mass production. However, there are several major limitations of mechanical cutting methods which can cause the failure of components due to the extreme shear stress at the contact point and the tool wear. To overcome these problems, the laser cutting method has been studied because it has many advantages that provide non-contact, flexibility, high-intensity energy, rapid manufacturing, and various applications [1,2,3,4]. Recently, the laser cutting of electrodes for lithium-ion batteries has been conducted. Lee et al. addressed that multiphysics modeling and the simulation of single-mode fiber laser were studied. Then, the optimal process parameters for cutting the lithium-ion batteries were found through various experiments [5,6,7,8]. Lutey et al. found the main factors affecting laser cutting efficiency and the quality of lithium-ion batteries [9,10]. Schmider et al. suggested the analytical model of the laser ablation of the lithium-ion battery [11]. In addition, laser cutting technology is applied to the carbon fiber reinforced plastic (CFRP) [12,13,14,15,16] and steel [17,18,19,20].

The spring contact probe (SCP) is widely used in semiconductor industries due to advantages including a high strength, fatigue resistance, and corrosion resistance. In addition, it is mainly manufactured by a stamping process, and it can improve thermal conductivity and electrical performance by coating with Au and Ni. However, the stamping process can cause deformation of the product due to mechanical stress and deteriorate the quality of the product by tool wear. To overcome these problems, the laser spot cutting methods have been used to manufacture the SCP for semiconductor package inspection. Although micro-laser cutting methods are being applied in the semiconductor industry, the low-quality cut finish, which affects the productivity, has not been completely solved. To avoid it, optimal laser cutting conditions are absolutely necessary. However, few studies have been conducted to analyze the micro-laser cutting technique for the SCP. Therefore, in this study, a series of laser spot cutting experiments are performed for Au-coated specimen. The Au-coating effect can improve the electrical performance of the component and prevent the wear and corrosion. Additionally, it has a high-quality cut finish due to its high thermal conductivity. Figure 1 shows the schematic of the laser spot cutting process. There is the difference between laser cutting and laser spot cutting. In the case of the laser cutting, there is a relative motion between laser and material during the machining process. On the other hand, in the case of the laser spot cutting, it can separate over two kinds of the part without a relative motion. To obtain appropriate process parameters, correlation analyses between the laser and coated material were performed. Then, appropriate laser spot cutting parameters were chosen for improving the quality of the component.

## 2. Experimental Setup and Design

As can be seen in Figure 2, the schematic diagram of the laser cutting system is shown. The laser source is the nano-second pulsed Ytterbium fiber laser (IPG-YLPM) with 1064 nm wavelength. The maximum average laser power (20 Watts) was used to maximize productivity. The laser was connected to a computer-controlled 3-D galvo scanner (RAYLASE AS-12Y) to project laser patterns with a high speed up to 10 m/s. The laser was focused on the specimen surface with 30 µm beam diameters, which has a Gaussian intensity profile. Figure 3 illustrates the specimen which was coated with gold with 4 µm thickness and nickel with 0.5 µm thickness on both sides of the Beryllium-Copper (BeCu) plate. Thus, the total thickness of the specimen was 54 µm. In addition, to investigate the interaction characteristics between the laser and Au-coated BeCu plate, a rectangular specimen was used.

A nano-second pulse laser, which has a relatively longer pulse duration, induces heat damage around the machined area, melting the affected volume. To improve process quality, therefore, parameters with less of a Heat-Affected Zone (HAZ) and burr formation are investigated. Table 1 lists various parameters in the experiments. The effect of energy transmission time was observed by changing pulse duration (Δ*t*) from 4 ns to ~200 ns. The repetition rate is inversely proportional to the pulse duration. For each pulse duration and repetition rate, various numbers of pulses were selected to have the same total irradiated energy (*E_total_*), then increased *E_total_* from 1 mJ to 1000 mJ. The total irradiated energy (mJ) is expressed by the following Equation (1):(1)$$E_{\mathrm{total}}=E\times N$$

In the above Equation (1), N is the number of pulses, and E is the pulse energy per one pulse (µJ). After machining is finished, the machined surface was observed by using a Scanning Electron Microscope (SEM). The size of HAZ and the top and bottom parts of Material Removal Zone (MRZ) were measured. Then, the roundness of Au-coated and non-coated specimens with different pulse durations was evaluated. Lastly, the surface morphology of a laser-irradiated area in accordance with pulse duration was observed on the Au-coated surface of specimens.

## 3. Experimental Results and Analysis

### 3.1. Analysis of HAZ and MRZ Results

The HAZ, MRZ top, and MRZ bottom were measured for each total irradiated energy, and each result is plotted in Figure 4, Figure 5 and Figure 6, respectively. In general, these sizes increase in proportion to *E_total_*, but the amount of change is slow under 10 mJ. Furthermore, the increasing rate is high over 10 mJ in the short pulse duration group (4, 8, 20 ns) while the longer duration group (50, 100, 200 ns) shows a slight increase.

In Figure 4, compared to results without coating [21], HAZ with Au-coating becomes smaller at 8, 20, 200, 800 mJ (*E_total_*) when pulse durations are 4, 8, 20, 50 ns, respectively. As can be seen in Figure 5, similarly, the MRZ top sizes between the coating and non-coating specimens were analyzed. The top-sized MRZ with the Au-coating case becomes smaller than the non-coating case at 8, 20, 80, 200 mJ with respect to 4, 8, 20, 50 ns. There were no significant changes in top HAZ and MRZ at a higher pulse duration (100, 200 ns). These results show that the Au-coated zone spreads energy quicker, which results in slightly bigger holes in the heat-affected zone at low total energy. However, as total energy increases, the energy spread to the copper specimen becomes dominant that it generates less HAZ and MRZ top than the non-coating case.

In Figure 6, we measured MRZ bottom and compared them to non-coating data. In the Au-coating case at 8 and 20 ns pulse duration, similar to top of MRZ, the dimension of bottom increases drastically from ober 100 mJ. In addition, the hole is penetrated at low total energy from 1 to 5 mJ. Due to the Au layer under the specimen, the energy is transferred at the bottom, which helps to penetrate the specimen. Thus, It can be confirmed that the thermal conductivity increased due to Au-coating, and the laser energy absorbed in specimen was transferred to the bottom without significant energy loss.

### 3.2. HAZ and MRZ Change Rate Analysis

The change rate of HAZ and MRZ was analyzed with regard to different pulse durations in Figure 7. ΔHAZ and ΔMRZ are defined as an Equations (2) and (3) below.
ΔHAZ = HAZ_max_ − HAZ_min_(2)
ΔMRZ = MRZ_max_ − MRZ_min_(3)

The result shows less variation at a high pulse duration. In this case, HAZ and MRZ are proportional to the total irradiated energy and pulse duration, having enough time to transfer heat energy to the material, which guarantees steady material removal. However, the low pulse duration group at 4, 8, and 20 ns shows more HAZ and MRZ variation, which means that the machining process is sensitive to total irradiated energy (*E_total_*). As you see in previous graphs, this group shows an abrupt increase in HAZ and MRZ at the high *E_total_* region, which causes an increase in the variation. Another aspect of the result is that the Au-coated case shows lower HAZ and MRZ variation in the low pulse duration group due to the faster heat spread on the Au-coated layer, which stabilizes the process. Compared to the non-coated specimen, the values of top and bottom MRZ for the Au-coated specimen are observed to be almost similar. Especially, in the low pulse duration group, variation in MRZ for the non-coated specimen is significant. Therefore, Au-coating is essential to achieve the uniform quality of the product by the laser spot cutting.

### 3.3. Roundness

In Figure 8, the roundness of the Au-coated and non-coated specimen with different pulse durations was illustrated. Roundness for each specimen was defined as Equation (4) below.
(4)$$Roundness=\frac{R_{min}}{R_{max}}\times 100\left(\%\right)$$

Roundness results of the Au-coated specimen were considered by our previous study [21]. Among them, the unmeasurable roundness and values exceeding the number of pulses were excluded. At Δ*t* = 200 ns, the roundness of the Au-coated and non-coated state was consistent. At Δ*t* = 100 ns, the roundness of the Au-coated and non-coated state was reduced to *E_Total_* = 100 mJ or more. At Δ*t* = 50 ns, the roundness of the Au-coated state was decreased to *E_Total_* = 100 mJ or more. However, in the non-coated state, the roundness was reduced to *E_Total_* = 200 mJ or more. At Δ*t* = 20 ns, in the coated state, the roundness was decreased to *E_Total_* = 80 mJ or more. In the non-coated state, the roundness was reduced to *E_Total_* = 500 mJ or more. At Δ*t* = 8 ns, in the coated state, the roundness was decreased to *E_Total_* = 500 mJ or more. In the non-coated state, the roundness was reduced to *E_Total_* = 200 mJ or more. The number of pulses increases as the total irradiated energy increases, resulting in excessive thermal effects. For this reason, this results in irregular material removal areas around the laser irradiation area, which causes a reduction in roundness.

### 3.4. Surface Morphology

Figure 9 presents SEM images of holes produced on the surface of Au-coated and non-coated specimens by laser spot cutting using a total irradiated energy of 10 mJ. In Figure 9a, a relatively wide splash mark and a bit of burr were observed on the surface. On the contrary, a large amount of burr formed by the re-solidification of materials that have not been fully removed was observed around the hole, as shown in Figure 9b. Compared to the non-contacted specimen, Au has the lower melting temperature and higher thermal conductivity, causing laser energy to transmit toward the bottom part, readily. Therefore, less molten material was distributed around the surface of the hole. In contrast, Ni that makes up the non-coated surface has the higher melting point and lower melting temperature. Thus, most laser energies were mainly absorbed by the surface, resulting in a large amount of burr around the hole. Figure 10 shows the SEM image of a top view of the laser-irradiated area on the Au-coated surface. In Figure 10a, at Δ*t* = 200 ns, a big splash mark around the hole was found at *E_Total_* = 8 mJ or more. An interaction area between laser and material moved downward during the laser irradiation. At the same time. the melting area accompanying the splash motion moved downward, too. After finishing the laser interaction, the melted materials were re-solidified. Thus, the hole penetration was formed in the Au-coated specimen [22]. At *E_Total_* = 5 mJ or less, it had no penetration with the laser mark. In Figure 10b, at Δ*t* = 100 ns, it was smaller than the previous one. In addition, the roundness was decreased at *E_Total_* = 500 mJ or more because the equilibrium of force between the surface tension and recoil pressure was lost. Re-solidified layers around the hole were generated at *E_Total_* = 200 mJ or more. Especially, the laser surface melting and rapid solidification of the material were investigated because there was enough time to apply the next pulse. By this phenomenon, the non-circular hole occurred from the irregular surface condition during laser irradiation. Figure 10c,d shows the characteristics of the surface at Δ*t* = 50 ns and 20 ns, which formed thick burrs around the hole. At Δ*t* = 50 ns, there was no hole penetration at *E_Total_* = 2 mJ or less. In particular, at Δ*t* = 20 ns, there were long burrs around the hole at *E_Total_* = 800 mJ or more. Asymmetric burrs were also found at *E_Total_* = 5 mJ or more, whereas symmetric burrs were formed at *E_Total_* = 2 mJ or less. In Figure 10e, the condition of the surface was shown at Δ*t* = 8 ns. The hole of the wrinkled shape was found at *E_Total_* = 500 mJ or more. There were few burrs around the MRZ from *E_Total_* = 80 mJ to *E_Total_* = 200 mJ. Burrs were stacked in the middle of the hole. In addition, there were few grains surrounding HAZ from *E_Total_* = 8 mJ to *E_Total_* = 100 mJ. Based on these observations, it can be noticed that the big splash mark around the laser irradiation area was formed as the pulse duration increased in low total irradiated energy. In addition, comparatively big hole on the top surface occurred at a high pulse duration because of a long interaction time between laser and material. However, from more than 500 mJ, the bigger hole was formed by the increase in the number of shots at a low pulse duration. At the same time, the large amount of burr around the hole was generated by the incomplete removal of material which is melted during the laser irradiation in low pulse duration.

## 4. Conclusions

In this paper, laser machining characteristics of a Au-coated SCP specimen with different laser process parameters were studied. Then, the results were compared with a non-coated specimen to observe the Au-coating effect. After laser experiments, HAZ, MRZ, roundness, and surface morphology in accordance with total irradiated energy were investigated. Experiment results can be divided into two groups: long and short pulse durations. The first group had higher HAZ and MRZ dimensions with less variation and laser parameter changes. In contrast, the second group was sensitive to laser parameter change, showing a drastic change when the total irradiated energy was above 10 mJ. At a high energy, having many numbers of pulses than the long pulse duration group, the hole size becomes bigger. Although the laser-irradiated area at a pulse duration of 200 ns had a high dimension of HAZ and MRZ, it had less variation and comparatively uniform roundness compared to other pulse durations. In short, when the total irradiated energy and pulse duration were 10 mJ and 200 ns, it was most suitable parameter in laser spot cutting for the Au-coated specimen. Furthermore, in the Au-coated specimen, hole penetration was observed at a low pulse duration and low total energy. As the pulse duration decreased, the number of pulses increased due to a low pulse energy under the same total irradiated energy. By an increase in the number of pulses, the hole penetration of the specimen was formed because of a repetitive thermal accumulation of laser. Moreover, at Δ*t* = 8 ns and 20 ns, the roundness of over 60% was obtained using a low total energy which was at *E_Total_* = 10 mJ or less. However, above *E_Total_* = 10 mJ, an irregular shape in the laser irradiation area is observed because of the excessive laser energy. Based on these results, it was confirmed that Au-coating helps to cut the spring contact probe in terms of laser processing compared to the non-coating. In addition, the effect of laser parameters on the hole penetration was systematically studied. Through these experimental results, this research can be beneficial to investigate the influence of Au-coating during laser spot cutting for SCPs. However, in this study, the tendency of the burr in accordance with the total irradiated energy was not clearly observed under the same pulse duration. These results were hypothesized to have been varied by the surface condition of the Au-coated specimen. Furthermore, the material characteristics before and after laser irradiation are not considered. To improve the performance of laser spot cutting on SCPs further, it is thus planned to include the physical and chemical changes before and after laser irradiation for describing a mechanism of the burr formed during the laser spot cutting.

## Figures and Tables

**Figure 1 materials-14-03300-f001:**
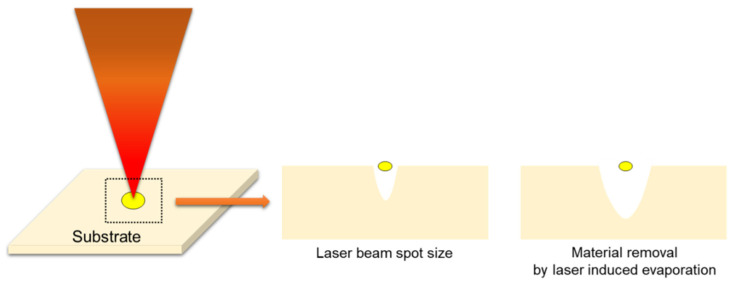
Schematic illustration of laser spot cutting process.

**Figure 2 materials-14-03300-f002:**
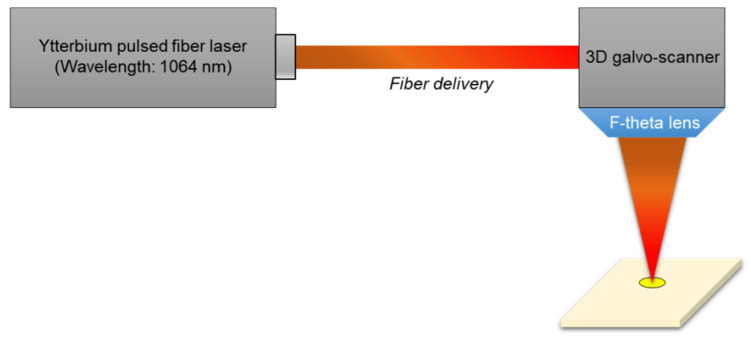
Schematic illustration of laser spot cutting system.

**Figure 3 materials-14-03300-f003:**
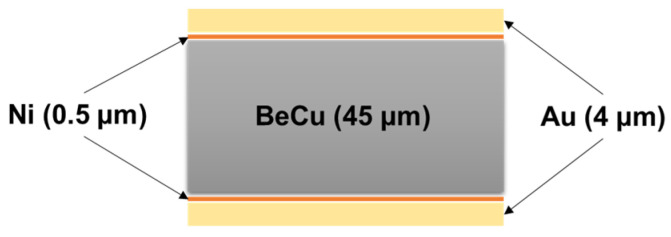
Beryllium-Copper specimen with metal coating.

**Figure 4 materials-14-03300-f004:**
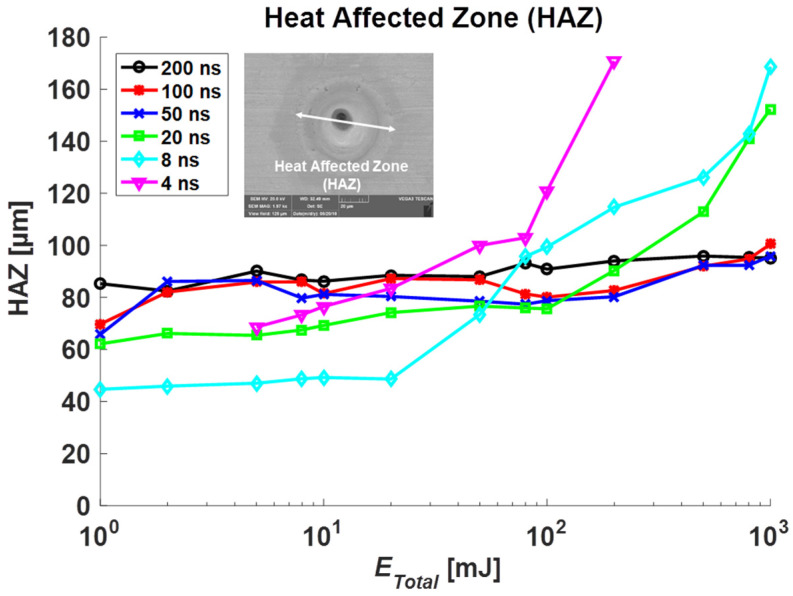
Measurement results in the Heat-Affected Zone (HAZ) for Au-coated specimens.

**Figure 5 materials-14-03300-f005:**
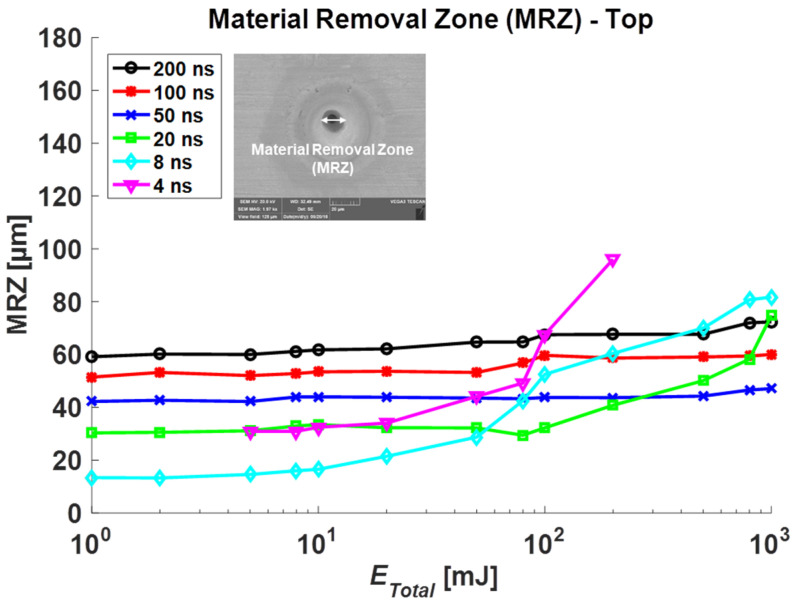
Measurement results in the top surface of Material Removal Zone (MRZ top) for Au-coated specimens.

**Figure 6 materials-14-03300-f006:**
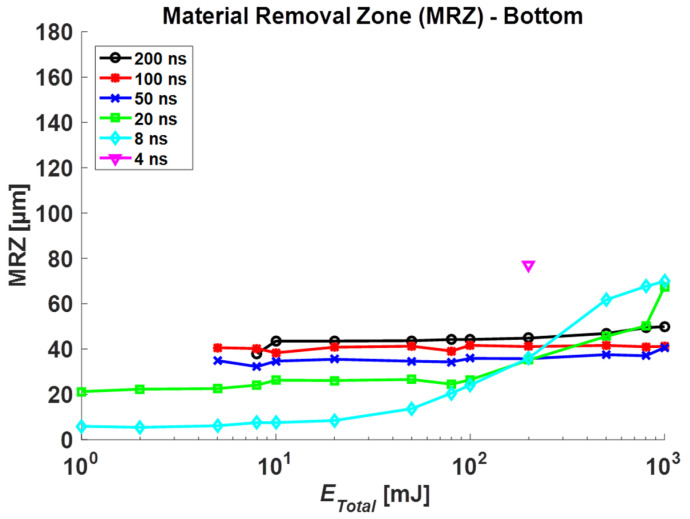
Measurement results in the bottom surface of Material Removal Zone (MRZ bottom) for Au-coated specimen.

**Figure 7 materials-14-03300-f007:**
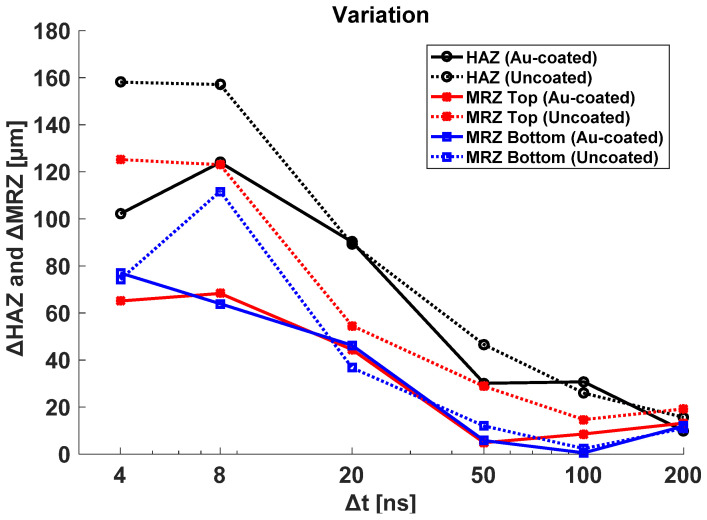
Variation of HAZ, MRZ top, and MRZ bottom with different pulse durations.

**Figure 8 materials-14-03300-f008:**
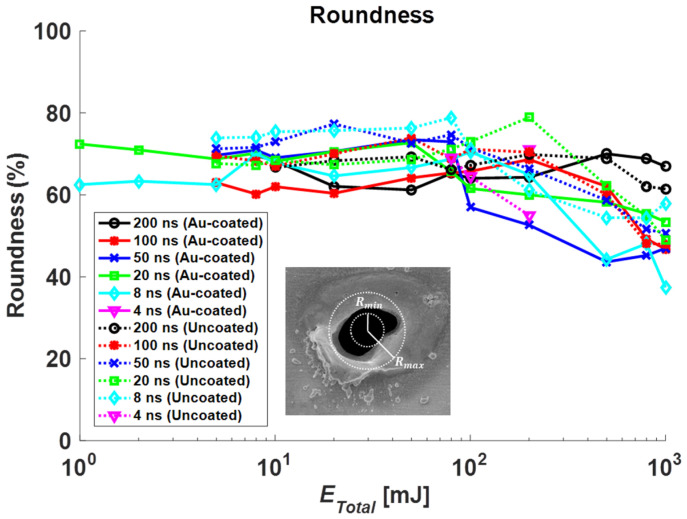
Roundness results under Au-coated and non-coated condition.

**Figure 9 materials-14-03300-f009:**
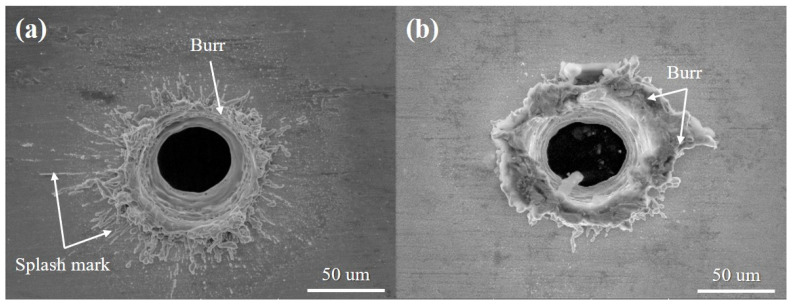
SEM images of holes in (**a**) Au-coated and (**b**) non-coated specimens formed by laser spot cutting (*E_Total_*
_=_ 200 mJ, Δ*t* = 200).

**Figure 10 materials-14-03300-f010:**
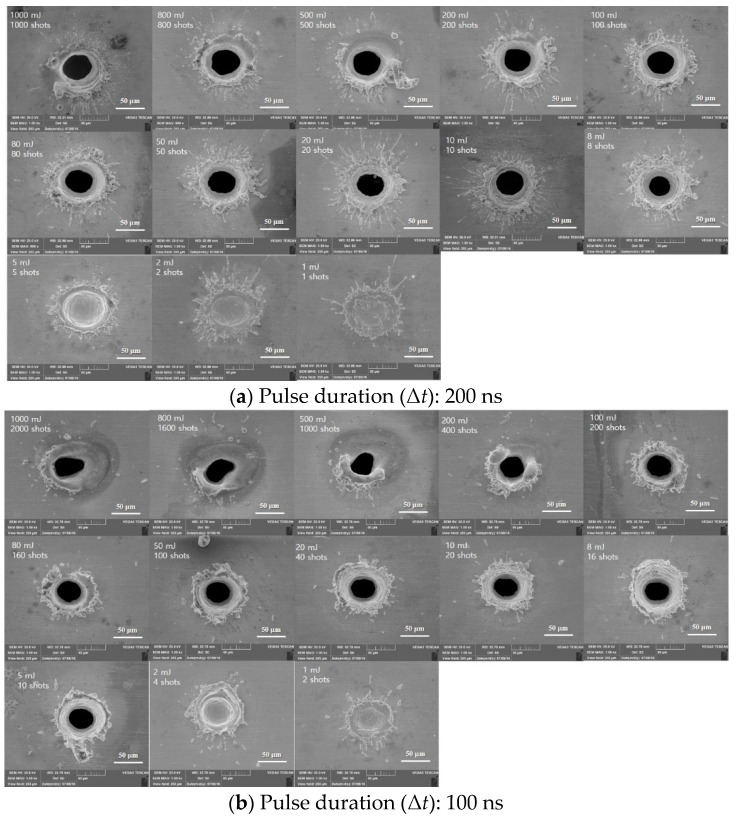
Top view of laser ablation on Au-coated surface with different pulse duration.

**Table 1 materials-14-03300-t001:** Experimental condition.

**Pulse Duration [ns]**	**Rep. Rate [kHz]**	**Pulse E [µJ]**	**Total E [mJ]**												
			1000	800	500	200	100	80	50	20	10	8	5	2	1
			**Number of Pulse [#]**												
200	20	1000	1000	800	500	200	100	80	50	20	10	8	5	2	1
100	40	500	2000	1600	1000	400	200	160	100	40	20	16	10	4	2
50	60	333.3	3000	2400	1500	600	300	240	150	60	30	24	15	6	3
20	105	190.5	5250	4200	2625	1050	525	420	263	105	53	42	26	11	5
8	200	100	10,000	8000	5000	2000	1000	800	500	200	100	80	50	20	10
4	500	40	25,000	20,000	12,500	5000	2500	2000	1250	500	250	200	125	50	25

## Data Availability

Data available in a publicly accessible repository.

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
