# Peer review of "Effect of Au-Coating on the Laser Spot Cutting on Spring Contact Probe (SCP) for Semi-Conductor Inspection"

_materials, 2021, doi:10.3390/ma14123300_

Round 1
Reviewer 1 Report
The authors present the laser spot cutting on Au-coating layer by a laser system. The proposed manuscript within the scope of the IJAMT journal. Some of the results observations are interesting, however, if the paper can be improved in the following areas, it would add more value to the readers:
a) For the industrial application, authors could provide the application fields (i.e. future size, critical properties, and etc.).
b) Abstract should contain results of the original works and should be summarized well.
c) The parameters selection should be systemically described.
d) For the experiments, the roundness evaluation, the evaluation formula is necessary.
e) The mechanism of laser spot formation and the non-circular hole phenomenons should be described.
f) Not only the morphology but also the contact electrical properties is critical for spring contact probe. In the manuscript without describe discussion.
Reviewer 2 Report
Thank you for your submission, the topic of the paper is very interesting. Micro-laser cutting is a very perspective method for this specific kind of problem. I have several comments and questions.
- On the third page is an error of the reference source.
- The results in figures 4, 5, and 6 are results from one experiment or it was made several times and the results are an average value?
- In line 86 is mentioned a comparison of the results in Fig. 4 with results for the uncoated version. But why are these results not given in the paper?
- It would be appropriate to supplement the results with generalized conclusions and possible evaluation of suitable parameters for the given application.
- How will the above results affect the change in the thickness of the Au-coating layer?
Reviewer 3 Report
The work concerns the influence of laser spot cutting parameters on the size of ablation zone, the heat affected zone and penetration depth of Au- coated samples
and without the coating. The topic of the work is interesting and important for application reasons.
Experimental works are of a basic nature and concern only the measurements of the indicated parameters at different laser spot cutting parameters (energy, pulse duration and repetition)
without a deeper analysis of the impact of laser processing parameters on changes in the microstructure of the cut elements, which may be significant due to their properties. The conclusions are imprecise, after reading the paper it is not clear whether the presence of Au coating has a positive effect on the cutting process or not. The working language is simple and clear. There are a few editing errors, one literature source (reference) is missing, as indicated in the .pdf file It is a promising work, but in the opinion of the reviewer it is required to improve the research quality by in-depth analysis of cutting areas microstructure in the cross-section, including EDS analysis. After completing the research, the manuscript may be re-submitted for publication in Materials

Round 2
Reviewer 2 Report
Thank you for the made improvement. Mainly the answer to my 4th question could be described in more detail and help to improve the scientific potential more. Did you see in my first review also the basic recommendation where I mentioned what can be or must be improved? Mainly the methods and results have to be better described?
Reviewer 3 Report
The manuscript text is supplemented and the discussion of the resultsis extended. Figures' description are corrected. The missing information
is entered as recommended by the reviewer but microstructural studies have not been performed.
There is no single image of the microstructure to show that the coating is present.
The work has the character of basic research, a kind of report
on the selection of laser cutting process parameters for coated and uncoated samples, due to the penetration depth and symmetry of the melting zone. research and in order to conduct an in-depth analysis of the impact of the above-mentioned parameters
on the microstructure and material properties changes in HAZ after laser cutting. Due to the scientific value of the manuscript, I do not recommend it for publication in Materials.
Author Response
Thanks for your comments.
